# RETHINKING SHAPLEY VALUE
# FOR DATA CONTRIBUTION

## ABSTRACT

Shapley value is a principled and widely used framework for data valuation in machine learning. However, its application has led to a critical, yet often overlooked, conceptual confusion between the value of a data point (its average utility across all subsets) and its specific, structural contribution (its role in shaping the final model). This conflation is problematic since valuation scores that are strongly influenced by small subsets may not reliably indicate the true contribution of a data point. To resolve this, we propose a framework designed to directly measure structural contribution. Our method modifies the Shapley formulation by 1) using a similarity-based utility function to capture impact on the global model structure, and 2) applying a Beta-weighting scheme to prioritize larger, more stable subsets. Experiments on SVMs show our method more accurately identifies support vectors, which serve as the ground truth for contribution, outperforming standard Shapley-based approaches in both precision and recall. This approach also shows strong performance in data pruning tasks and is applicable to broader probabilistic models. Our work provides not just a new method, but a clearer conceptual framework to distinguish the valuation of a data point from its true contribution.

## 1 INTRODUCTION

Quantifying the importance of individual data points is a fundamental challenge in modern machine learning, underpinning applications from robust data selection (Wang et al., 2024) and targeted data pruning (Yang et al., 2023) to data markets (Liu et al., 2021a;b; Zhang et al., 2024). This endeavor, commonly known as data valuation, has increasingly relied on the Shapley value from cooperative game theory as a principled framework for assigning value (Shapley, 1953; Ghorbani & Zou, 2019; Jia et al., 2019; Rozemberczki et al., 2022; Xu et al., 2023; Wang et al., 2025).

However, the application of this framework has revealed a critical but overlooked conceptual ambiguity, which conflates the value of a data point with its true contribution. The Shapley value of a point reflects its average marginal utility across coalitions, whereas its contribution denotes its specific, structural role in shaping the final model. These notions need not coincide, and treating them as equivalent risks obscuring the true drivers of model behavior. This paper takes the position that valuation and contribution are fundamentally distinct. Much of the existing literature implicitly equates the two, but we argue that this conflation is both conceptually flawed and empirically misleading. Our goal is therefore not only to propose a new algorithm but also to establish a clearer framework for understanding what Shapley values truly measure.

The gap between valuation and contribution stems from the fact that the effect of a data point in machine learning is highly context-dependent. Unlike in classical cooperative games, where adding a new player usually improves or at least preserves utility, the influence of a point in ML can vary significantly: it may help, harm, or have no effect at all depending on the subset of data used for training. The Shapley value captures the average of these volatile effects, while the contribution of a data point is determined by its role in the final full dataset. This fundamental difference leads to significant, practical misinterpretations of the true role of a data point. For instance, a redundant data point may garner a high Shapley value because it is helpful across many small, unstable subsets. However, its true contribution is negligible, as its function is entirely supplanted by similar examples in the full dataset. Conversely, a point that is critical for the final decision boundary might be penalized in numerous arbitrary coalitions because small subsets often fail to learn the correct

boundary, resulting in a low or even negative value. This score, while perhaps a fair assessment of its volatile average impact, fails to capture its indispensable contribution to the final model.

We construct two toy coalition games (Table 1) to illustrate this gap. These examples highlight a sharp conflict between the calculated Shapley value and what is intuitively understood as contribution. In Example 1, the negative score of player C can be interpreted as a fair valuation of its average disruptive effect, yet this valuation masks its positive final contribution to the grand coalition. In Example 2, player C receives a positive valuation, even though its participation clearly harms the final group outcome by lowering the total utility from 60 to 50. This paradox illustrates the danger of conflating valuation with contribution. A valuation, by averaging utility across all possible scenarios, can fail to capture the contribution of a point on the final model. This misinterpretation is especially problematic in structured models like Support Vector Machines (SVMs), where true contribution is sparse and determined as a support vector.

Table 1: Two illustrative examples distinguishing Valuation (the Shapley value) from Contribution (impact on the grand coalition).

| Coalition | Example 1 Utility | Example 2 Utility |
|---|---|---|
| $\{A\}$ | 20 | 30 |
| $\{B\}$ | 20 | 30 |
| $\{C\}$ | $-10$ | 20 |
| $\{A, B\}$ | 50 | 60 |
| $\{A, C\}$ | 15 | 60 |
| $\{B, C\}$ | 15 | 60 |
| $\{A, B, C\}$ | 60 | 50 |
| $\mathcal{SV}_C$ | $-1.67$ | $+5.00$ |

The SVM model (Cortes & Vapnik, 1995) provides an ideal setting to distinguish valuation from contribution, since its architecture clearly separates support vectors, which define the decision boundary, from non-support vectors, which play no structural role. This setting offers an unambiguous ground truth for contribution. An effective contribution metric must therefore prioritize these critical points. Yet the standard Shapley value fails this test(see our SVM experiments in Section 5.2), often assigning high scores to non-support vectors due to their incidental utility in small subsets, thereby misrepresenting true contribution.

To directly measure contribution, we propose a new framework that modifies the standard Shapley value in two main ways. The key intuition is that the true contribution of a point is determined by its impact on the final model, rather than its average performance in small, unstable subsets. First, we redefine the utility of a subset. Instead of using a simple performance score like accuracy, we measure how similar a model trained on the subset is to the final model trained on all data. This similarity score directly captures the influence of a data point on the structure of the model, such as the shape of its decision boundary. Second, we reweight coalition sizes to emphasize large, stable subsets near the full dataset. This prioritization yields a more faithful measure of contribution by focusing on the role of a point in near-complete models.

Our contributions are threefold:

- We highlight the problems that arise from confusing these two ideas in machine learning.

- We propose a new method to directly measure contribution that more accurately identifies support vectors, the ground truth for contribution, than the standard Shapley value.

- We demonstrate strong practical utility in data pruning, where removing low-contribution points identified by our method leads to smaller accuracy degradation compared to baselines.

By disentangling valuation from contribution, our framework provides a clearer view of how data shapes machine learning models. It shifts the focus from average performance in small subsets to structural impact on the full model, offering a principled and scalable approach to data contribution analysis. Beyond assigning values, our method opens new directions for understanding and leveraging data in diverse models.

## 2 RELATED WORK

Our work builds on the growing literature on data valuation in machine learning. Shapley values have been widely adopted as a principled tool for attributing utility to individual data points (Ghorbani & Zou, 2019; Jia et al., 2019; Rozemberczki et al., 2022). While effective for assigning holistic valuations, these methods typically conflate the average marginal utility of a point with the structural role of a point in the final model. Alternative approaches include influence functions (Cook, 1977; Koh & Liang, 2017; Pruthi et al., 2020; Bae et al., 2022). However, influence functions are essentially a local approximation to leave-one-out retraining. They measure sensitivity to the removal of a single point but do not account for interactions among data points. This issue is especially pronounced in settings such as SVMs, where only a sparse set of points contributes decisively to the decision boundary.

Beyond the standard uniform-weighted Shapley formulation, several weighted variants have been proposed to prioritize subsets of particular sizes. For instance, Beta Shapley (Kwon & Zou, 2022) assigns larger weights to smaller coalitions in order to mitigate noise sensitivity. In contrast, our framework applies a complementary strategy: we reweight toward larger, nearly complete coalitions. We discuss the synergies and differences between our weighting mechanism and that of Beta Shapley in Appendix E. Due to space limits, we discuss these and more related works thoroughly in Appendix B.

## 3 PRELIMINARIES

### 3.1 SUPPORT VECTOR MACHINES (SVM)

Support Vector Machines (SVMs) are a class of supervised learning models used for classification and regression tasks. Given a labeled dataset $\{(x_i, y_i)\}_{i=1}^n$, where $x_i \in \mathbb{R}^d$ and $y_i \in \{-1, +1\}$, the standard linear SVM solves the following optimization problem:

$$\min_{w,b} \frac{1}{2}\|w\|^2 \quad \text{s.t.} \quad y_i(w^\top x_i + b) \geq 1, \ \forall i.$$

In practice, the soft-margin SVM is commonly used to handle non-separable data:

$$\min_{w,b,\xi} \frac{1}{2}\|w\|^2 + C\sum_{i=1}^n \xi_i \quad \text{s.t.} \quad y_i(w^\top x_i + b) \geq 1 - \xi_i, \ \xi_i \geq 0.$$

The solution is typically expressed in terms of a subset of training points, known as support vectors, which lie on or violate the margin boundaries and have non-zero dual coefficients. The decision function takes the form:

$$f(x) = \text{sign}\left(\sum_{i=1}^n \alpha_i y_i K(x_i, x) + b\right),$$

where $K(\cdot, \cdot)$ is a kernel function, and $\alpha_i$ are dual variables obtained from the corresponding dual optimization problem. Only support vectors contribute to the prediction, making SVMs a natural setting for studying data point contributions and instance-level interpretability.

### 3.2 SEMIVALUE FRAMEWORK FOR DATA VALUATION

A **semivalue** extends the Shapley construct by weighting average marginal utility of a player with any non-negative vector $\boldsymbol{\omega} = \langle \omega_1, \ldots, \omega_N \rangle$ satisfying $\sum_{j=1}^N \omega_j = 1$ (Carreras et al., 2003). For a dataset $D$ of size $N$ and utility $\mathcal{U}$, define the average marginal contribution of point $i$ to coalitions of size $j - 1$ as

$$\Delta_j(i; \mathcal{U}) = \frac{1}{\binom{N-1}{j-1}} \sum_{\substack{S \subseteq D \setminus \{i\} \\ |S| = j-1}} \left[\mathcal{U}(S \cup \{i\}) - \mathcal{U}(S)\right]. \tag{1}$$

The semivalue of $i$ is then given by:

$$\mathcal{V}_i(\mathcal{U}) = \sum_{j=1}^N \omega_j \, \Delta_j(i; \mathcal{U}). \tag{2}$$

The classical **Shapley value** uses uniform weights $\omega_j^{\mathrm{Sh}} = \frac{1}{N}$, the unique choice that satisfies the *efficiency* axiom, $\sum_i V_i = \mathcal{U}(D)$. The **Banzhaf value** instead weights all coalitions equally, resulting in weights $\omega_j^{\mathrm{Bz}} = \binom{N-1}{j-1} \cdot \frac{1}{2^{N-1}}$ that prioritize mid-sized coalitions. Efficiency is relaxed in the Banzhaf value in favor of other properties.

Following nearly all prior Data Shapley work, we take the *validation-set accuracy* to be the default choice of utility function and denote it by $\mathcal{U}_{\mathrm{acc}}$. For a model $\hat{y}_S$ trained on subset $S$, the utility is

$$\mathcal{U}_{\mathrm{acc}}(S) \;=\; \frac{1}{|D_{\mathrm{val}}|} \sum_{(x,y)\,\in\, D_{\mathrm{val}}} \mathbf{1}\big\{\hat{y}_S(x) = y\big\}. \tag{3}$$

Unless stated otherwise, all baselines tagged "acc" use $\mathcal{U}_{\mathrm{acc}}$.

## 4 METHODOLOGY

We propose a data valuation framework that redefines both the utility function and the Shapley aggregation process. The framework aims to capture the contribution of each data point, rather than relying solely on predictive accuracy.

### 4.1 DESIGN MOTIVATION: FROM GOAL TO PRINCIPLES

To ground our design choices, we first ask: what properties should an ideal measure of *contribution* possess?

**Principle 1: Structural fidelity.** Contribution should reflect the role of a data point in shaping the *global structure* of the learned model. Therefore, the utility function cannot be restricted to coarse outcomes such as accuracy on subsets. It must capture how closely a subset-trained model aligns with the full model's decision function.

**Principle 2: Robustness to small, unstable coalitions.** The key distinction between valuation and contribution lies in the volatility introduced by tiny subsets. An effective aggregation scheme must down-weight these noisy cases and place greater emphasis on large, representative coalitions where structural effects become reliable.

These principles directly motivate our framework: in Section 4.2 we introduce similarity-based utilities that align subset and full-model behavior, and in Section 4.3 we develop a Beta-weighted aggregation that emphasizes large subsets.

### 4.2 SIMILARITY-BASED UTILITY FUNCTIONS

Guided by Principle 1, we redefine the utility function to capture how a subset-trained model aligns with the full model. In Shapley-based data valuation, the utility function $\mathcal{U}(S)$ is typically defined as prediction accuracy on a validation set(Eq. 3), but this metric is coarse, binary, and insensitive to fine-grained model behavior (Xia et al., 2024). To more faithfully reflect how closely a subset-trained model approximates the full model, we propose defining utility as a similarity score between the two models. This similarity-based utility function serves as a soft structural alignment metric and is central to our framework.

**RKHS-based Functional Similarity (SVM-specific).** We begin with a theoretically grounded similarity measure based on the Reproducing Kernel Hilbert Space (RKHS) norm. Suppose $f_S$ and $f_{\mathrm{full}}$ denote the decision functions of SVMs trained on subset $S$ and the full dataset, respectively. Let $\mathcal{H}$ be the RKHS induced by a kernel function $k(\cdot, \cdot)$. We define the utility of subset $S$ as the cosine similarity between the two models in $\mathcal{H}$:

$$\mathcal{U}_{\mathrm{RKHS}}(S) = \frac{\langle f_S, f_{\mathrm{full}} \rangle_{\mathcal{H}}}{\|f_S\|_{\mathcal{H}} \cdot \|f_{\mathrm{full}}\|_{\mathcal{H}}}.$$

This formulation naturally aligns with the functional view of SVMs, where the solution lies in a Hilbert space defined by kernel evaluations (Schölkopf & Smola, 2002). It emphasizes agreement in both decision direction and margin structure. However, this approach relies heavily on dual

representations and kernel machinery, making it specific to kernelized models like SVMs. It is not directly applicable to non-kernel models such as tree ensembles or neural networks. To address this limitation, we next introduce a model-agnostic similarity utility based on probabilistic outputs.

**KL-based Predictive Similarity (Model-agnostic).** To extend the utility function to a broader class of models, we propose a predictive similarity metric based on probabilistic outputs. Let $p_j^{\text{full}}$ and $p_j^{\text{subset}}$ be the predicted class probability vectors on validation point $x_j$ from the full and subset models, respectively. We define the utility of $S$ as:

$$\mathcal{U}_{\text{KL}}(S) = \exp\left(-\frac{1}{|D_{\text{val}}|} \sum_{x_j \in D_{\text{val}}} \frac{\text{KL}(p_j^{\text{full}} \| p_j^{\text{subset}}) + \text{KL}(p_j^{\text{subset}} \| p_j^{\text{full}})}{2}\right),$$

where $\text{KL}(\cdot \| \cdot)$ denotes the Kullback–Leibler divergence. This symmetric KL-based similarity reflects how well the subset model matches the predictive behavior of the full model across the entire distribution, following principles of f-divergence estimation (Nguyen et al., 2010). A utility value close to 1 indicates near-identical predictive distributions, while values near 0 indicate significant divergence.

To ensure numerical stability, we apply smoothing with a small constant $\varepsilon > 0$ to avoid undefined log terms:

$$\text{KL}(p \| q) = \sum_c p(c) \log \frac{p(c) + \varepsilon}{q(c) + \varepsilon}.$$

Additionally, in cases where the subset model degenerates to a single-class predictor—common for small or skewed subsets—we explicitly define $\mathcal{U}_{\text{KL}}(S) = 0$ to reflect maximum dissimilarity.

**Intuition for KL Similarity** Symmetric KL divergence offers a balanced measure of model similarity by averaging forward and reverse KL, thereby mitigating the mode-missing and mode-covering biases of the asymmetric variants. More importantly, it is sensitive to fine-grained shifts in class probabilities, allowing it to distinguish models that achieve identical accuracy but yield different probability distributions (e.g., 0.9/0.1 vs. 0.6/0.4). This continuity makes $\mathcal{U}_{\text{KL}}$ a more faithful proxy for decision function behavior, particularly when subsets are small or degenerate.

**Why similarity-based utility matters.** Unlike accuracy, which only reflects top-1 agreement, $\mathcal{U}_{\text{KL}}$ preserves information about the full probability vector. This property makes it more indicative of how each data point shapes the decision boundary. We now formalize this advantage with a local sensitivity theorem and a lemma on the piecewise-constant nature of accuracy.

**Theorem 4.1** (Local quadratic sensitivity of KL similarity). *Assume Assumptions C.1, C.2, C.3 hold. Let $P$ be the orthogonal projection onto $(\ker J_V)^\perp$. There exists $\varepsilon > 0$ such that for all $\delta\theta$ with $\|\delta\theta\| \le \varepsilon$,*

$$1 - \mathcal{U}_{\text{KL}}(\theta^\star + \delta\theta) = \tfrac{1}{2}\,\delta\theta^\top G\,\delta\theta + o(\|\delta\theta\|^2),$$

*and*

$$\frac{m}{2}\,\|P\delta\theta\|^2 \ \le \ 1 - \mathcal{U}_{\text{KL}}(\theta^\star + \delta\theta) \ \le \ \frac{M}{2}\,\|P\delta\theta\|^2.$$

**Lemma 4.2** (Accuracy is locally constant inside a margin ball). *Let $\gamma_{\min} = \min_{x \in D_{\text{val}}}\big(z_{\theta^\star}^{(1)}(x) - z_{\theta^\star}^{(2)}(x)\big) > 0$ be the smallest logit gap on the validation set, and let $G_{\max} = \max_{x \in D_{\text{val}}} \|\nabla_\theta f_{\theta^\star}(x)\|$. Then for all $\|\delta\theta\| < \gamma_{\min}/G_{\max}$ we have $\text{Acc}(\theta^\star + \delta\theta) = \text{Acc}(\theta^\star)$.*

We provide complete proofs of Theorem 4.1 and Lemma 4.2 in Appendix C.1. This result highlights that $\mathcal{U}_{\text{KL}}$ is not only more generalizable but also more sensitive to the true influence of data points, especially near decision boundaries. As we show later in experiments, this improved fidelity allows our method to more precisely identify structurally critical points such as SVM support vectors.

### 4.3 BETA-WEIGHTED SEMIVALUE AGGREGATION

As argued in Section 4.1, our second design principle, robustness to small coalitions, requires a weighting scheme that reduces the influence of unstable small subsets and prioritizes larger, more representative ones. While many functional forms could achieve this effect, we adopt the Beta distribution for three main reasons. First, it offers high flexibility: by adjusting its two parameters

$(\alpha, \beta)$, one can recover uniform weighting (standard Shapley), emphasize small coalitions (as in Beta Shapley (Kwon & Zou, 2022)), or emphasize large coalitions (our choice). Second, this family enables direct comparison with prior work, positioning our formulation as a natural counterpart to Beta Shapley. Third, the Beta distribution has convenient mathematical properties: its closed-form normalization yields concise derivations when combined with combinatorial coefficients, avoiding the need for heuristic approximations.

Building on these reasons, we formally define our Beta-weighted framework. For a dataset of size $N$, let $\Delta_j(i)$ denote the average marginal contribution of data point $i$ to all subsets of size $j$. We define the Beta-weighted Shapley value as:

$$\mathcal{SV}_i^{\text{beta}} = \sum_{j=1}^{N} \omega_j^{\text{beta}} \cdot \Delta_j(i), \tag{4}$$

where the weights $\omega_j^{\text{beta}}$ are given by:

$$\omega_j^{\text{beta}} = \frac{\binom{N-1}{j-1} \cdot \text{Beta}(j + \beta - 1, N - j + \alpha)}{\text{Beta}(\alpha, \beta)}, \quad \text{with } \beta > \alpha.$$

By setting $\beta > \alpha$, this formulation biases the aggregation toward larger subsets $j \to N$, effectively discounting marginal contributions from small and potentially noisy coalitions. Such a bias is particularly desirable when the utility function rewards structural similarity to the full model, which only manifests reliably in larger subsets.

**Relation to Beta Shapley.** Beta Shapley (Kwon & Zou, 2022) defines a parametric family of *semivalues* by placing a $\text{Beta}(\alpha, \beta)$ distribution over coalition sizes and averaging marginal contributions accordingly. It recovers the classical Shapley value at $(\alpha, \beta) = (1, 1)$, and commonly adopts settings that emphasize *small* coalitions (e.g., $\alpha > \beta$) to improve robustness and unify prior weighting schemes. Our formulation differs in both *goal* and *weighting regime*. We explicitly bias toward *large*, near-full coalitions ($\alpha < \beta$) and couple this aggregation with a similarity-based utility designed to capture *structural contribution* (decision-boundary fidelity), rather than subset-level accuracy. Both approaches are semivalues and thus, in general, do not satisfy the efficiency axiom. A side-by-side comparison of weighting, axiomatic properties, and empirical behavior appears in Appendix E.

We provide an illustration of how different Beta parameters affect the weighting scheme in Appendix G, where the weight curves of `Beta(1,1)`, `Beta(8,1)`, and `Beta(1,8)` are visualized.

It is important to note that by introducing Beta-distributed weights, the efficiency property, which guarantees that the sum of all data points' values equals the utility of the full dataset ($\sum \mathcal{SV}_i = \mathcal{U}(D)$), is no longer held in our framework. The sum of Beta-weighted Shapley values will not generally equal the total utility. This is a conscious trade-off: we sacrifice strict axiomatic efficiency to gain a more robust and insightful valuation that better aligns with a data point's true contribution, mitigating the misleading influence from small, unrepresentative coalitions.

**Why weighting improves estimation.** Standard (unweighted) Shapley aggregation treats all subset sizes equally, which may dilute the influence of data points that are only impactful in large, high-fidelity subsets. In contrast, by assigning higher weights to larger subsets, our Beta-weighted formulation places greater trust in marginal contributions that better reflect the full model behavior. We formalize this insight in the following result:

**Theorem 4.3.** *Suppose there exist two distinct data points $i, j \in N$, and constants $m, \epsilon > \delta$, satisfying for all subsets $S \subseteq N \setminus \{i, j\}$ with $|S| \geq m$:*

$$\mathcal{U}(S \cup \{i\}) - \mathcal{U}(S) \geq \epsilon, \quad \mathcal{U}(S \cup \{j\}) - \mathcal{U}(S) \leq \delta.$$

*Then, as long as the total weight assigned to large subsets ($|S| \geq m$) is sufficiently large, we have:*

$$\mathcal{SV}_i^{beta} > \mathcal{SV}_j^{beta}.$$

*Thus, weighted Shapley values emphasizing large subsets better differentiate consistently important data points.*

A full proof is provided in Appendix C.3. This result illustrates how weighting can correct the overvaluation of non-critical points in the standard Shapley formulation by focusing on contributions where structural importance becomes evident.

In practice, the exact computation of Eq. 4 is infeasible for large $N$. We therefore approximate it by Monte Carlo sampling over subset sizes and random coalitions. The full procedure is provided in Algorithm 1 in Appendix F.

## 5 EMPIRICAL VERIFICATION

We empirically evaluate our proposed data valuation framework along two key axes: (1) its ability to accurately identify structurally important data points, such as SVM support vectors, and (2) the effect of removing points identified as uninformative (i.e., having low absolute values). We compare our method against several baselines using benchmark datasets. While our primary focus is on SVMs, we also include an experiment using logistic regression with KL-based utility (Appendix D.2) to demonstrate the framework's extensibility beyond margin-based models.

### 5.1 DATASETS AND SETUP

We conduct experiments on five benchmark datasets with varying scales and structures. The **Iris** dataset is a classic low-dimensional dataset with well-understood structure, widely used in SVM analysis. The **Wine** dataset from UCI provides a medium-scale benchmark with clear class structure, which is suitable for evaluating robustness in structured classification tasks. The **Breast Cancer** dataset, also from UCI, is a widely used binary classification benchmark that reflects performance in medical-related structured data scenarios. The **Ionosphere** dataset, characterized by moderately high-dimensional features, is commonly employed to evaluate generalization ability in signal classification. Finally, the **CIFAR-10** dataset serves as a large-scale benchmark of natural images; instead of using raw pixel inputs, we employ a ResNet-based feature extractor to obtain 256-dimensional feature representations for our experiments.

For each dataset, we train an SVM classifier and compute data valuation scores using the following methods[1]:

- `Beta(1,1)-acc`: Standard Shapley Value with accuracy-based utility.
- `Beta(8,1)-acc`: Original Beta Shapley in paper (Kwon & Zou, 2022).
- `Beta(1,8)-kl/RKHS`: Beta-weighted Shapley with similarity-based utility.
- `Banzhaf value`: Marginal contribution under uniform coalition weights (Wang & Jia, 2023).

### 5.2 SUPPORT VECTOR IDENTIFICATION

We first assess the ability of each method to identify structurally important points, specifically SVM support vectors. Figure 1 shows sorted Shapley value distributions for the Iris dataset, where each bar corresponds to a data point and red bars indicate true support vectors. The standard Shapley (Beta(1,1)-acc) assigns a high value to many non-support vectors, resulting in a confusing outcome that obscures the distinction between critical and irrelevant points. In contrast, our KL-based method with Beta(1,8) weighting produces a sharper separation, allocating high value to support vectors while suppressing irrelevant points.

We evaluate support vector identification from two complementary perspectives. First, we treat the task as a ranking problem and measure performance using Precision@K%, Recall@K%, and AUC, with results summarized in Table 2. Second, for a more fine-grained analysis, we directly measure the correlation (Pearson $r$ and Spearman $\rho$) between valuation scores and the SVM dual coefficient $|\alpha_i|$. A detailed correlation analysis, provided in Appendix D.3, further supports these findings.

Table 2 summarizes support vector identification across all datasets, reporting Precision@5%/20%, Recall@5%/20% (relative to training set size), and AUC scores that capture overall discriminative

---

[1]Kwon & Zou identify $(\alpha, \beta) = (16,1)$ as optimal for data valuation, whereas Tamine et al. (Tamine et al., 2025) use (4,1) in the benchmarks. We adopt (8,1) as a compromise and include (1,8) for comparison.

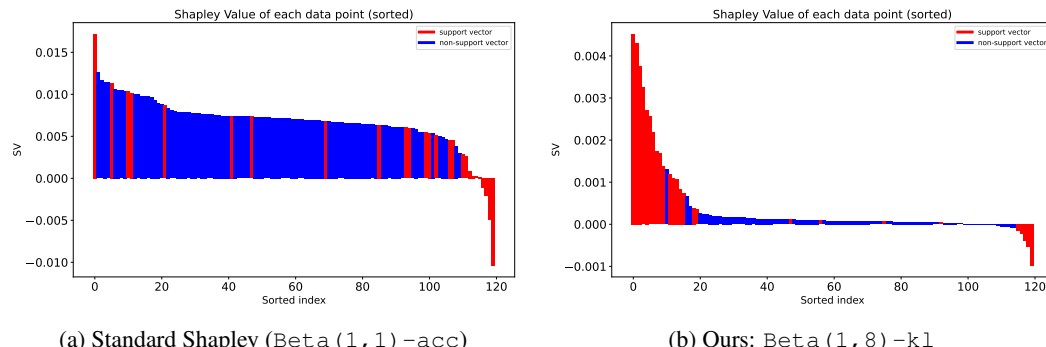

(a) Standard Shapley (`Beta(1,1)-acc`)      (b) Ours: `Beta(1,8)-kl`

Figure 1: Sorted Shapley values on the Iris dataset, with support vectors marked in red. **Left:** Standard Shapley with accuracy utility overemphasizes non-critical points. **Right:** Our method (`Beta(1,8)-kl`) concentrates value on support vectors, achieving better structural alignment.

power. Across all datasets, our methods, especially `Beta(1,8)-rkhs`, consistently outperform the standard accuracy-based baseline. Notably, `Beta(1,8)-rkhs` achieves near-perfect Precision@5% on all datasets and even perfect Precision@20% on Iris, demonstrating strong structural alignment with the true support vectors.

Table 2: Support Vector Identification Performance

| Dataset | Method | Precision@5% | Recall@5% | Precision@20% | Recall@20% | AUC |
|---|---|---|---|---|---|---|
| Iris | Beta(1,1)-acc | 0.667 | 0.154 | 0.292 | 0.269 | 0.324 |
| | Banzhaf value | **1.000** | **0.231** | 0.458 | 0.423 | 0.546 |
| | Beta(1,8)-kl | **1.000** | **0.231** | 0.708 | 0.654 | 0.712 |
| | Beta(1,8)-rkhs | **1.000** | **0.231** | **0.958** | **0.885** | **0.987** |
| Wine | Beta(1,1)-acc | 0.286 | 0.095 | 0.143 | 0.191 | 0.375 |
| | Banzhaf value | 0.714 | 0.238 | 0.357 | 0.477 | 0.539 |
| | Beta(1,8)-kl | **1.000** | **0.333** | **0.607** | **0.810** | **0.917** |
| | Beta(1,8)-rkhs | **1.000** | **0.333** | **0.607** | **0.810** | 0.816 |
| Breast Cancer | Beta(1,1)-acc | 0.409 | 0.250 | 0.121 | 0.306 | 0.341 |
| | Banzhaf value | 0.773 | 0.472 | 0.231 | 0.583 | 0.597 |
| | Beta(1,8)-kl | 0.864 | 0.528 | 0.319 | 0.806 | 0.839 |
| | Beta(1,8)-rkhs | **1.000** | **0.611** | **0.374** | **0.944** | **0.945** |
| Ionosphere | Beta(1,1)-acc | 0.571 | 0.107 | 0.375 | 0.280 | 0.366 |
| | Banzhaf value | 0.786 | 0.147 | 0.554 | 0.413 | 0.475 |
| | Beta(1,8)-kl | **1.000** | **0.187** | 0.875 | 0.653 | 0.824 |
| | Beta(1,8)-rkhs | **1.000** | **0.187** | **1.000** | **0.747** | **0.896** |
| CIFAR-10 | Beta(1,1)-acc | **1.000** | **0.058** | 0.930 | 0.214 | 0.527 |
| | Banzhaf value | **1.000** | **0.058** | 0.990 | 0.228 | 0.587 |
| | Beta(1,8)-kl | **1.000** | **0.058** | **1.000** | **0.230** | 0.931 |
| | Beta(1,8)-rkhs | **1.000** | **0.058** | **1.000** | **0.230** | **0.941** |

## 5.3 REMOVAL BY ASCENDING ABSOLUTE VALUE

To further assess the quality of the semivalue-based valuations, we conduct a point removal experiment, deleting points from the lowest to the highest absolute semivalue $|\mathcal{V}_\rangle|$. The goal is to evaluate whether points assigned low absolute semivalues are indeed uninformative and can be safely removed without degrading model performance.

For each method, we sort all training data points by the absolute value of their score in ascending order. We then incrementally remove points and retrain the SVM classifier from scratch at each step, measuring the test accuracy. An effective valuation method should maintain high accuracy even after a large fraction of points (those with near-zero scores) have been removed.

Figure 2 presents the results of this experiment on three datasets (Iris, Ionosphere, and Breast Cancer). Across all datasets, our similarity-based method (`Beta(1,8)-kl`) demonstrates remarkable robustness: test accuracy remains nearly unchanged until a large fraction of the training set is removed (over 80% in the case of Iris), indicating that the valuations successfully concentrate model importance on a compact critical subset. By contrast, accuracy-based baselines such as `Beta(1,1)-acc` and `Beta(8,1)-acc` deteriorate much earlier overall. This indicates that these methods incorrectly assign low absolute scores (near-zero importance) to some critical data points, causing them to be removed prematurely. And the Banzhaf value, with its emphasis on mid-sized coalitions, also demonstrates better robustness than standard Shapley, reinforcing that the choice of coalition weighting is critical. These findings consistently highlight the structural fidelity of our similarity-aware, large-subset-weighted approach across diverse datasets.

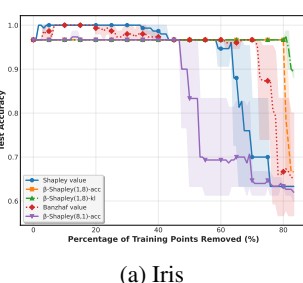 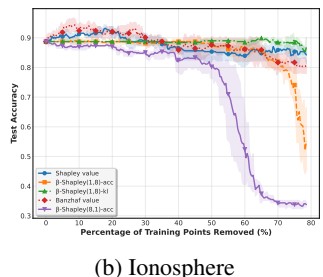 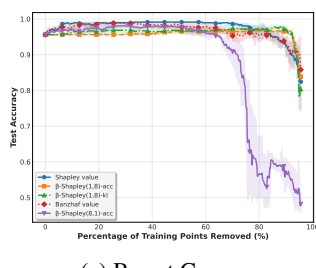

| (a) Iris | (b) Ionosphere | (c) Breast Cancer |

Figure 2: Point removal experiment based on absolute semivalue on three datasets. (a) Iris, (b) Ionosphere, and (c) Breast Cancer. At each step we remove the lowest absolute semivalue points and retrain the SVM. Our method `Beta(1,8)-kl` preserves test accuracy until a large fraction of data is removed (over 80% on Iris; see main text), while accuracy-based baselines degrade much earlier.

## 5.4 DISCUSSION

These results confirm that our proposed Beta-weighted, similarity-based Shapley framework more faithfully captures the structural contribution of data points. It successfully highlights critical points like support vectors and downweights incidental or redundant samples, aligning with the theoretical motivations discussed in earlier sections.

**Extending to Deep Models** While current experiments focus on SVMs (and logistic regression in Appendix D.2), our similarity-based utility, especially the KL-divergence variant, is broadly applicable to any probabilistic model, including deep neural networks, as it operates on output distributions. Nonetheless, subset retraining in neural networks can be prohibitive, and adapting our framework in conjunction with tools such as neural tangent kernel (NTK) theory or other functional similarity metrics remains an important direction for future work.

## 6 CONCLUSION AND LIMITATIONS

In this work, we re-examined the value of data by distinguishing between its average utility (valuation) and its structural role in shaping models (contribution). We introduced a framework that combines similarity-based value functions with Beta-weighted aggregation, enabling a direct assessment of structural importance. Experiments across multiple datasets confirmed its effectiveness at identifying points with high structural contribution, such as support vectors in SVMs. This perspective not only provides a sharper language for data value but also opens new directions for understanding and managing data in learning systems.

A key limitation is that our most detailed analysis and empirical results currently center on SVMs and small-scale models; while the KL-based utility is, in principle, model-agnostic, further theoretical and large-scale empirical validation on deep neural networks remains future work, especially since subset retraining in neural networks could be prohibitive.

ETHICS STATEMENT

This study uses only public benchmark datasets, including Iris, Wine, Breast Cancer, Ionosphere, and CIFAR-10, with experiments on CIFAR-10 conducted on 256-dimensional ResNet features rather than raw images. No personal data or sensitive attributes are involved, and use follows the original dataset terms. No private or sensitive data is involved. The methods are designed for research purposes and do not raise foreseeable ethical concerns.

REPRODUCIBILITY STATEMENT

All datasets, model configurations, and experimental details are described in Section 5 and the appendices. Code and scripts for reproducing the results are included in the anonymous repository provided with the submission.

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

## A  USE OF LARGE LANGUAGE MODELS

In accordance with ICLR policy on large language model usage, we report the role of LLMs in the preparation of this manuscript. We employed GPT-5 as a writing assistance tool for grammar, clarity, and stylistic polishing. We also used GitHub Copilot for code completion and small boilerplate in the IDE. The authors reviewed and tested all code. The tool did not design methods or implement core algorithms. The LLM did not design methods, implement algorithms in a decisive way, analyze data, or interpret results.

The LLM was not used for research ideation, design of methods, data analysis, or interpretation of results. All scientific ideas, experimental designs, theoretical results, and conclusions presented in this paper are entirely the work of the authors.

# B EXTENDED RELATED WORK

**Shapley-based Data Valuation**  The Shapley value has emerged as a principled method for quantifying the contribution of individual data points in machine learning, drawing from cooperative game theory. Classical formulations, such as Data Shapley (Ghorbani & Zou, 2019), estimate marginal utility across all subsets, typically using validation accuracy as the utility function. Several approximation algorithms have been developed to make Shapley estimation feasible in large-scale settings (Jia et al., 2019). However, these methods assume equal importance across subset sizes and are sensitive to noise in small coalitions, potentially misrepresenting a point's global importance.

Unlike classical *convex* (super-additive) cooperative games—where rational agents *self-select* coalitions only when their payoff does not decrease—training data are *passively* aggregated by the practitioner. Thus the characteristic function $\mathcal{U}(\cdot)$ need not be monotone: adding a mislabeled or redundant example can even lower utility, i.e. $\mathcal{U}(S \cup \{i\}) < \mathcal{U}(S)$. This structural mismatch explains the negative or counter-intuitive marginal contributions often observed in data-Shapley studies (see Appendix E).

**Semivalue Variants**  Beyond the standard uniform-weighted Shapley value, several works have explored semivalue formulations that reweight subsets of different sizes. The Banzhaf value (Wang & Jia, 2023) assigns equal weight to all coalitions, thereby emphasizing medium-sized subsets at the expense of efficiency, reflecting the intuition that such subsets carry more balanced information than very small or very large ones. More recently, Beta Shapley (Kwon & Zou, 2022) introduces a flexible Beta distribution over subset sizes, with a particular focus on smaller coalitions to mitigate noise sensitivity. In contrast, our framework applies a complementary strategy: we reweight toward larger, nearly complete subsets, where model behavior is most stable and representative. If one accepts that reweighting by subset size is beneficial—as argued by both Banzhaf and Beta Shapley—then emphasizing large coalitions is a natural next step, especially when utility is defined via model-level similarity. This perspective guides our framework, which prioritizes near-complete subsets to provide a more faithful measure of structural contribution.

**Utility Functions Beyond Accuracy**  While most Shapley-based data valuation methods define utility in terms of predictive accuracy, recent work has questioned the sufficiency of this coarse metric. For instance, P-Shapley (Xia et al., 2024) modifies the utility function to evaluate expected gains in predicted probabilities, thereby softening the decision boundary and improving sensitivity in probabilistic classifiers. However, it still fundamentally measures prediction-level outcomes. Wang et al. (Wang et al., 2025) propose a single-run estimation method that avoids retraining by using a differentiable surrogate utility based on leave-one-out gradients. In contrast, our framework redefines utility in terms of structural similarity between models, either through RKHS functional alignment or symmetric KL divergence over output distributions. The former builds on classical RKHS theory for functional comparison (Schölkopf & Smola, 2002), while the latter follows the principles of f-divergence estimation using symmetric KL, as formalized in (Nguyen et al., 2010). This shift enables a more faithful reflection of a data point's influence on the decision function, particularly in structured models like SVMs, where only a subset of points shape the boundary.

**Other Data Valuation Approaches**  In addition to Shapley-based methods, other valuation strategies include influence functions (Koh & Liang, 2017), leave-one-out (Cook, 1977). While these approaches offer alternative perspectives, they often lack formal axiomatic justification or encounter scalability bottlenecks, particularly in structured models. Our framework addresses these limitations by offering a principled and efficient valuation strategy grounded in cooperative game theory and model similarity.

In contrast to existing approaches, our framework integrates similarity-aware utility functions with a theoretically grounded Beta-weighted Shapley aggregation. This dual refinement allows us to better identify structurally critical data points (e.g., SVM support vectors), which may be overlooked by traditional formulations.

## C  COMPLETE PROOF OF THEOREM

### C.1  PROOF OF THEOREM 4.1

We restate the standing assumptions used in Theorem 4.1 and then give a concise proof.

**Assumption C.1** (Smooth logits). *For every validation input $x$, the logit map $z(x; \theta) \in \mathbb{R}^C$ is twice continuously differentiable in a neighbourhood of $\theta^\star$. Write $J(x) = \left.\frac{\partial z(x;\theta)}{\partial \theta}\right|_{\theta = \theta^\star}$ and stack them vertically to form $J_V \in \mathbb{R}^{(|D_{\mathrm{val}}| C) \times d}$.*

**Justification.** This holds for the models we study (SVMs, logistic regression) where logits are affine in parameters, hence $C^\infty$. For neural networks with piecewise-linear activations (e.g., ReLU), $z(x; \theta)$ is piecewise $C^\infty$ and second-order expansions hold almost everywhere; if needed one can adopt standard smooth surrogates (e.g., Softplus, GELU) or mollify logits, which leaves the empirical results unchanged but satisfies the technical smoothness required for a local Taylor expansion.

**Assumption C.2** (Nondegenerate validation probabilities). *There exists $\gamma \in (0, \frac{1}{2})$ such that for all $x \in D_{\mathrm{val}}$ and classes $c$, $p_{\theta^\star}(x)_c \in [\gamma, 1 - \gamma]$. Let $F(x) = \mathrm{Diag}(p_{\theta^\star}(x)) - p_{\theta^\star}(x) p_{\theta^\star}(x)^\top$. Then $F(x)\mathbf{1} = 0$, and there exist constants $0 < c_\gamma \le C_\gamma < \infty$ such that for all $v \perp \mathbf{1}$,*

$$c_\gamma \|v\|^2 \ \le \ v^\top F(x)\, v \ \le \ C_\gamma \|v\|^2.$$

**Justification.** The condition simply rules out saturated probabilities (0 or 1) that make KL curvature unbounded. It is standard in practice and can be enforced numerically by *temperature scaling* or a tiny *probability smoothing* $\varepsilon$ (both already used when computing $\mathcal{U}_{\mathrm{KL}}$). Hence it is a benign technical assumption ensuring the Fisher matrix $F(x)$ is well-conditioned on the validation set.

**Assumption C.3** (Fisher weighted identifiability). *Let*

$$G := \frac{1}{|D_{\mathrm{val}}|} \sum_{x \in D_{\mathrm{val}}} J(x)^\top F(x) J(x).$$

*Then $G$ is positive definite on the observable parameter subspace $\mathrm{Im}(J_V^\top) = (\ker J_V)^\perp$, with spectrum contained in $[m, M]$ for some $0 < m \le M < \infty$.*

**Justification.** We do not require full-rank identifiability in the entire parameter space; the theorem is stated on the observable parameter subspace $\mathrm{Im}(J_V^\top) = (\ker J_V)^\perp$. This matches standard identifiability assumptions in GLMs and ensures that parameter directions that actually affect validation logits are not degenerate. In practice, one can verify this by checking that the smallest singular value of the empirical matrix $G$ (or of $J_V$) is bounded away from zero; when architectural symmetries make some directions unidentifiable, the subspace restriction explicitly excludes them.

*Proof.* Let $\theta = \theta^\star + \delta\theta$ with $\|\delta\theta\|$ small. By Assumption C.1, for each $x$ the logit increment admits the Taylor expansion

$$\Delta z(x) := z(x; \theta) - z(x; \theta^\star) = J(x)\, \delta\theta + R_1(x), \qquad \|R_1(x)\| = O(\|\delta\theta\|^2).$$

The symmetric KL at $x$ admits a second-order expansion around $\theta^\star$ (the softmax is smooth and its Hessian at $\theta^\star$ equals $F(x)$ in logit coordinates):

Let $p^\star = \mathrm{softmax}(z^\star)$, where $z^\star := z(x; \theta^\star)$, $q(\Delta z) = \mathrm{softmax}(z^\star + \Delta z)$, and $\phi(z) = \log \sum_{i=1}^C e^{z_i}$. Then

$$\mathrm{KL}\big(p^\star \| q(\Delta z)\big) = \sum_{i=1}^C p_i^\star \big( \log p_i^\star - \log q_i(\Delta z) \big)$$

$$= -p^{\star\top} \Delta z \ + \ \phi(z^\star + \Delta z) - \phi(z^\star),$$

$$\mathrm{KL}\big(q(\Delta z) \| p^\star\big) = \sum_{i=1}^C q_i(\Delta z) \big( \log q_i(\Delta z) - \log p_i^\star \big)$$

$$= q(\Delta z)^\top \Delta z \ - \ \big( \phi(z^\star + \Delta z) - \phi(z^\star) \big).$$

Hence the symmetric KL at a fixed $x$ satisfies

$$\delta(x) = \tfrac{1}{2}\Big[\mathrm{KL}\big(p^\star\|q(\Delta z)\big) + \mathrm{KL}\big(q(\Delta z)\|p^\star\big)\Big] = \tfrac{1}{2}\big(q(\Delta z) - p^\star\big)^\top \Delta z.$$

The Jacobian of the softmax at $z^\star$ equals

$$J_{\mathrm{softmax}}(z^\star) = \mathrm{Diag}(p^\star) - p^\star p^{\star\top} =: F(x),$$

so a first–order expansion gives

$$q(\Delta z) = p^\star + F(x)\,\Delta z + O(\|\Delta z\|^2).$$

Substitution yields the second–order expansion

$$\delta(x) = \tfrac{1}{2}\,\Delta z^\top F(x)\,\Delta z \; + \; O(\|\Delta z\|^3).$$

Averaging over $D_{\mathrm{val}}$ and using $\mathcal{U}_{\mathrm{KL}}(\theta) = \exp\big(-\tfrac{1}{|D_{\mathrm{val}}|}\sum_x \delta(x)\big)$ yields, for some constant $K > 0$,

$$1 - \mathcal{U}_{\mathrm{KL}}(\theta) = \frac{1}{|D_{\mathrm{val}}|}\sum_x \delta(x) + O\!\left(\Big(\tfrac{1}{|D_{\mathrm{val}}|}\sum_x \delta(x)\Big)^2\right)$$

$$= \frac{1}{2|D_{\mathrm{val}}|}\sum_x \Delta z(x)^\top F(x)\,\Delta z(x) + O(\|\delta\theta\|^3),$$

where all big-$O$ constants are uniform over $x$ in a neighborhood of $\theta^\star$, so averaging preserves the $O(\|\delta\theta\|^3)$ rate.

Substituting $\Delta z(x) = J(x)\delta\theta + R_1(x)$ and absorbing the cubic remainder gives

$$1 - \mathcal{U}_{\mathrm{KL}}(\theta^\star + \delta\theta) = \frac{1}{2|D_{\mathrm{val}}|}\sum_x (J(x)\delta\theta)^\top F(x)(J(x)\delta\theta) + O(\|\delta\theta\|^3).$$

Let $w = J_V\,\delta\theta$ and $F_V := \mathrm{blkdiag}\big(F(x) : x \in D_{\mathrm{val}}\big)$. Then

$$1 - \mathcal{U}_{\mathrm{KL}}(\theta^\star + \delta\theta) = \frac{1}{2|D_{\mathrm{val}}|}\sum_{x \in D_{\mathrm{val}}} \big(J(x)\delta\theta\big)^\top F(x)\big(J(x)\delta\theta\big) \; + \; O(\|\delta\theta\|^3)$$

$$= \frac{1}{2|D_{\mathrm{val}}|}\,w^\top F_V\,w \; + \; O(\|\delta\theta\|^3)$$

$$= \tfrac{1}{2}\,\delta\theta^\top G\,\delta\theta \; + \; O(\|\delta\theta\|^3),$$

By Assumption C.3, there exist $m, M > 0$ and $\varepsilon > 0$ such that, for all $\delta\theta \in (\ker J_V)^\perp$ with $\|\delta\theta\| \le \varepsilon$,

$$\tfrac{m}{2}\,\|\delta\theta\|^2 \; \le \; 1 - \mathcal{U}_{\mathrm{KL}}(\theta^\star + \delta\theta) \; \le \; \tfrac{M}{2}\,\|\delta\theta\|^2,$$

where the $O(\|\delta\theta\|^3)$ term has been absorbed into the quadratic term by choosing a small enough neighborhood.

$\square$

## C.2  PROOF OF LEMMA 4.2

Let $f_\theta(x) = g_{y(x)}(x) - \max_{c \neq y(x)} g_c(x)$ denote the multiclass margin. By definition of $\gamma_{\min}$, every validation point satisfies $|f_{\theta^\star}(x)| \ge \gamma_{\min}$. Let $G_{\max} = \max_{x \in D_{\mathrm{val}}} \|\nabla_\theta f_{\theta^\star}(x)\|$. For any perturbation with $\|\delta\theta\| < \gamma_{\min}/G_{\max}$, the mean-value bound gives $|f_{\theta^\star + \delta\theta}(x) - f_{\theta^\star}(x)| \le G_{\max}\|\delta\theta\| < \gamma_{\min}$, so the sign of every margin is preserved. Therefore the $\arg\max$ labels on $D_{\mathrm{val}}$ do not change and the validation accuracy remains constant, i.e. $\mathrm{Acc}(\theta^\star + \delta\theta) = \mathrm{Acc}(\theta^\star)$. $\square$

## C.3    PROOF OF THEOREM 4.3

**Theorem 4.3 (Large subset dominance, restated).**    Let $D = \{1, \ldots, N\}$ be the data universe and $\mathcal{U}(\cdot)$ any utility function. Pick two distinct points $i, j \in D$. Assume there exist an integer $m$ and constants $\epsilon > \delta$ such that for every subset $S \subseteq D \setminus \{i, j\}$ with $|S| \geq m$

$$\mathcal{U}(S \cup \{i\}) - \mathcal{U}(S) \ \geq \ \epsilon, \qquad \mathcal{U}(S \cup \{j\}) - \mathcal{U}(S) \ \leq \ \delta.$$

Let $\mathcal{SV}_\ell^\beta$ be the Beta weighted Shapley value in Section 4.3, with weights $\{\omega_t^\beta\}_{t=1}^N$ satisfying $\omega_t^\beta \geq 0$ and $\sum_{t=1}^N \omega_t^\beta = 1$. Define

$$W_{\text{large}} := \sum_{t=m}^N \omega_t^\beta, \qquad M := \max_{1 \leq t < m} \left| \Delta_t(i) - \Delta_t(j) \right| \ \text{(with } M = 0 \text{ if } m = 1\text{)}.$$

If

$$W_{\text{large}} > \frac{M}{\epsilon - \delta + M},$$

then $\mathcal{SV}_i^\beta > \mathcal{SV}_j^\beta$.

*Proof.* Decompose

$$\mathcal{SV}_i^\beta - \mathcal{SV}_j^\beta = \underbrace{\sum_{t=1}^{m-1} \omega_t^\beta \left[ \Delta_t(i) - \Delta_t(j) \right]}_{S_{\text{small}}} + \underbrace{\sum_{t=m}^N \omega_t^\beta \left[ \Delta_t(i) - \Delta_t(j) \right]}_{S_{\text{large}}}.$$

**Large subsets.** For $t \geq m$, the assumption gives $\Delta_t(i) - \Delta_t(j) \geq \epsilon - \delta$, hence

$$S_{\text{large}} \ \geq \ (\epsilon - \delta)\, W_{\text{large}}.$$

**Small subsets.** For $t < m$, by the definition of $M$, $|\Delta_t(i) - \Delta_t(j)| \leq M$, hence

$$S_{\text{small}} \ \geq \ -M \, (1 - W_{\text{large}}).$$

**Combine.** Therefore

$$\mathcal{SV}_i^\beta - \mathcal{SV}_j^\beta \ \geq \ (\epsilon - \delta)\, W_{\text{large}} - M(1 - W_{\text{large}}) = (\epsilon - \delta + M)\, W_{\text{large}} - M,$$

which is positive whenever $W_{\text{large}} > \dfrac{M}{\epsilon - \delta + M}$. $\qquad\qquad\square$

# D  ADDITIONAL EXPERIMENT

## D.1  VISUALIZATION OF SHAPLEY HEATMAPS

To provide a comprehensive comparison, we visualize stacked Shapley heatmaps across all methods in Figure 3. Each row corresponds to a specific valuation method, and columns represent samples sorted by `Beta(1,1)-acc`. Support vectors are marked with "x". Our method (`Beta(1,8)-rkhs`) produces the most concentrated value on support vectors, while standard formulations such as `Beta(1,1)-acc` and `Beta(8,1)-acc` disperse credit across many less relevant points. This qualitative trend highlights the advantage of using similarity-based utility functions and size-aware weighting.

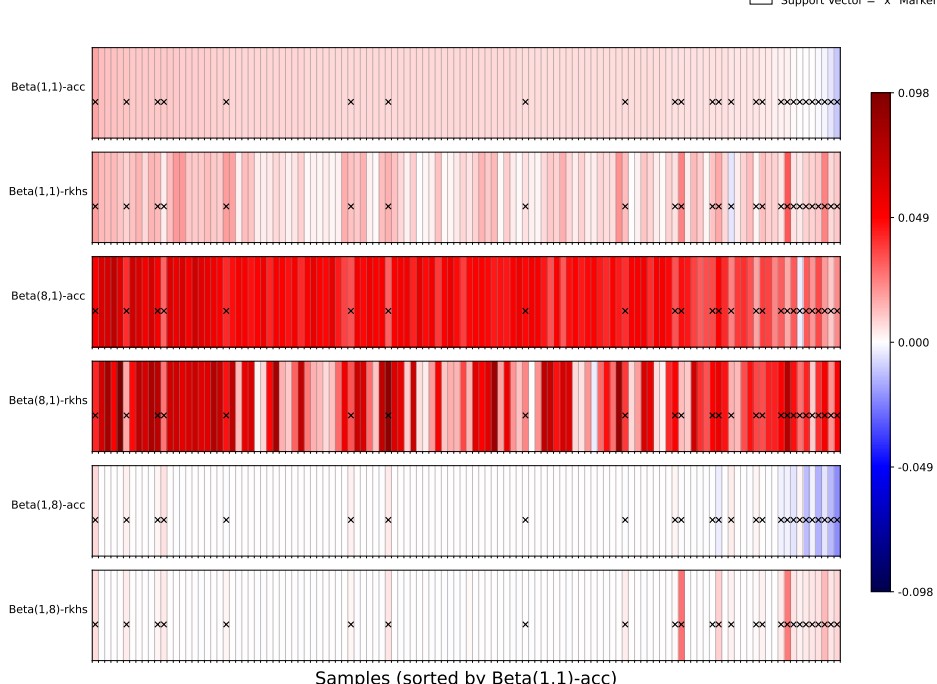

Figure 3: Stacked heatmaps of Shapley values under different weighting schemes and utility functions. Each row corresponds to a configuration (Beta parameters and utility), and samples are ordered by `Beta(1,1)-acc`. Support vectors are marked with "x". Our method (Beta(1,8)-rkhs) shows strong alignment between high Shapley values and true support vectors.

## D.2  LOGISTIC REGRESSION WITH KL-BASED UTILITY

To demonstrate the broader applicability of our framework beyond SVMs, we conduct an additional experiment using logistic regression on the Wine dataset. Unlike SVMs, logistic regression does not rely on support vectors, but still reflects decision boundary structure through its probabilistic outputs.

We apply the same Beta-weighted Shapley value estimation using KL-based utility (`Beta(1,8)-kl`) and compare it against standard accuracy-based Shapley (`Beta(1,1)-acc`) as well as additional variants (`Beta(1,1)-kl`, `Beta(8,1)-acc`, `Beta(8,1)-kl`, and Banzhaf). Since logistic regression outputs class probabilities, it provides a natural setting for KL-divergence-based model similarity.

**Setup.**  We train a logistic regression model using the same training-validation split as in the main experiments. For each data point, we compute Shapley values using both methods and perform a low-to-high deletion test: points with the lowest valuation scores are removed in increasing percentages, and the model is retrained after each removal.

**Results.**    As shown in Figure 4, our KL-based method (`Beta(1,8)-kl`) consistently outperforms all baselines, including standard accuracy-based Shapley (`Beta(1,1)-acc`), KL-based Shapley with alternative Beta parameters (`Beta(1,1)-kl`, `Beta(8,1)-kl`), accuracy-based Beta weighting (`Beta(8,1)-acc`), and Banzhaf. In particular, our method maintains stable test accuracy even after removing up to 80% of training points, whereas the performance of other methods—especially `Beta(1,1)-acc`—drops sharply once more than 40% of the data is removed.

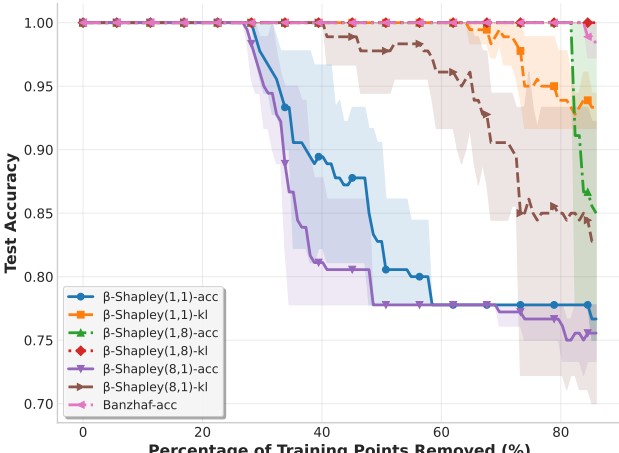

Figure 4: Low-to-high deletion experiment on Wine using logistic regression. KL-based valuation (Beta(1,8)-kl) yields more robust accuracy than standard Shapley (Beta(1,1)-acc).

**Conclusion.**    This experiment confirms that our KL-based, Beta-weighted framework is effective beyond structured models like SVMs. Even in models lacking explicit sparsity, such as logistic regression, our method identifies impactful training points more reliably than accuracy-based Shapley values.

### D.3    CORRELATION ANALYSIS WITH SVM DUAL COEFFICIENTS

To quantitatively evaluate how well each data valuation method captures the contribution of a point, we correlate the generated scores with the magnitude of the SVM dual coefficients, $|\alpha_i|$. The $|\alpha_i|$ values serve as a ground-truth measure of a point's influence on the learned decision boundary. A high positive correlation indicates that a valuation method successfully identifies structurally critical points.

The results on the Wine dataset (Figure 5) reveal a sharp divide. Accuracy-based methods, such as standard Shapley and Banzhaf values, show weak to negative correlation with the SVM coefficients (e.g., Pearson's $r = -0.265$ for Shapley). Visually, these methods produce an unstructured cloud of points, failing to distinguish support vectors from other data. In stark contrast, our similarity-based utilities demonstrate strong positive alignment. The KL-based method (Figure 5c) achieves a high correlation ($r = 0.762$), while the RKHS-based utility (Figure 5d) shows an exceptionally strong linear relationship ($r = 0.946$).

To ensure these findings are not limited to low-dimensional data, we replicate this analysis on the high-dimensional **CIFAR-10** ResNet features (Figure 6). The results are remarkably consistent. The accuracy-based baselines again show no meaningful correlation with $|\alpha_i|$. Conversely, our KL and RKHS similarity utilities maintain their strong positive correlation with the SVM boundary structure, achieving high Pearson coefficients of $r = 0.801$ and $r = 0.863$, respectively.

Across both datasets, these experiments confirm that similarity-based utilities consistently and effectively identify the data points that are decisive in shaping the model's final structure, a task where traditional accuracy-based valuations fall short.

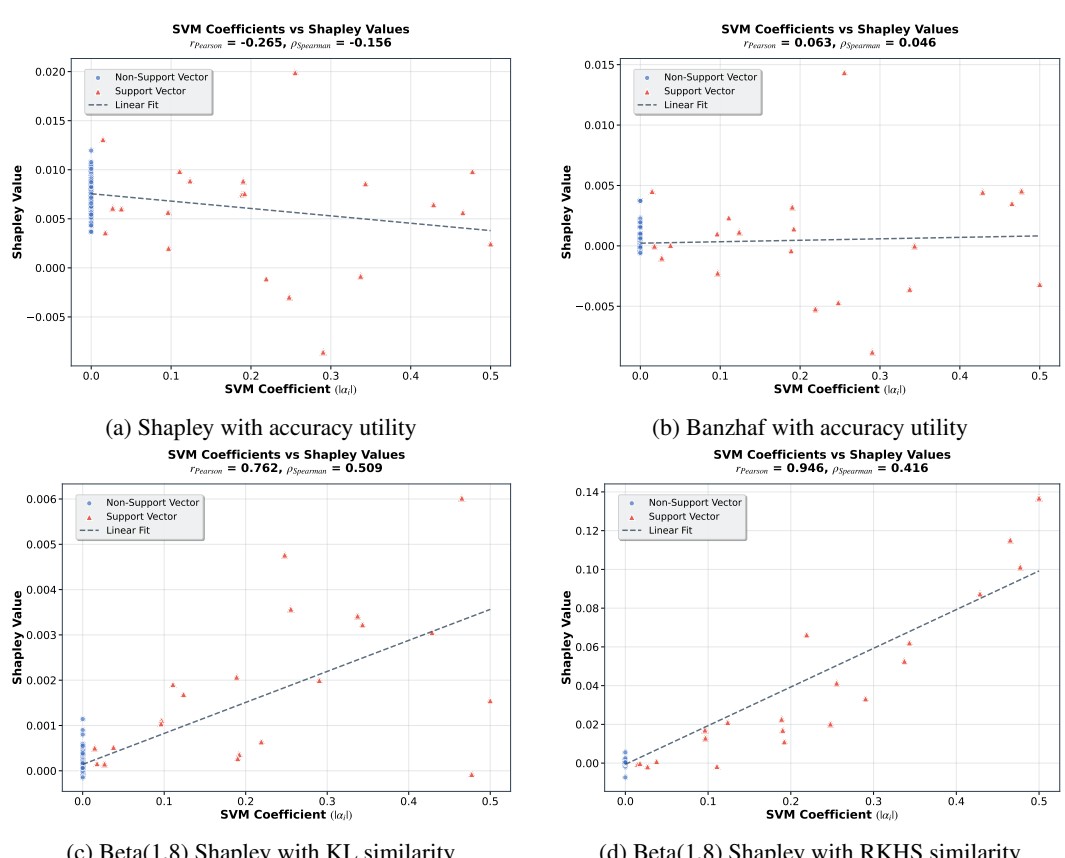

(a) Shapley with accuracy utility

(b) Banzhaf with accuracy utility

(c) Beta(1,8) Shapley with KL similarity

(d) Beta(1,8) Shapley with RKHS similarity

Figure 5: Correlation between $|\alpha_i|$ and data valuation scores on Wine. Each point is a training example; color indicates support vector membership. Similarity-based utilities align with the boundary more strongly than accuracy-based baselines.

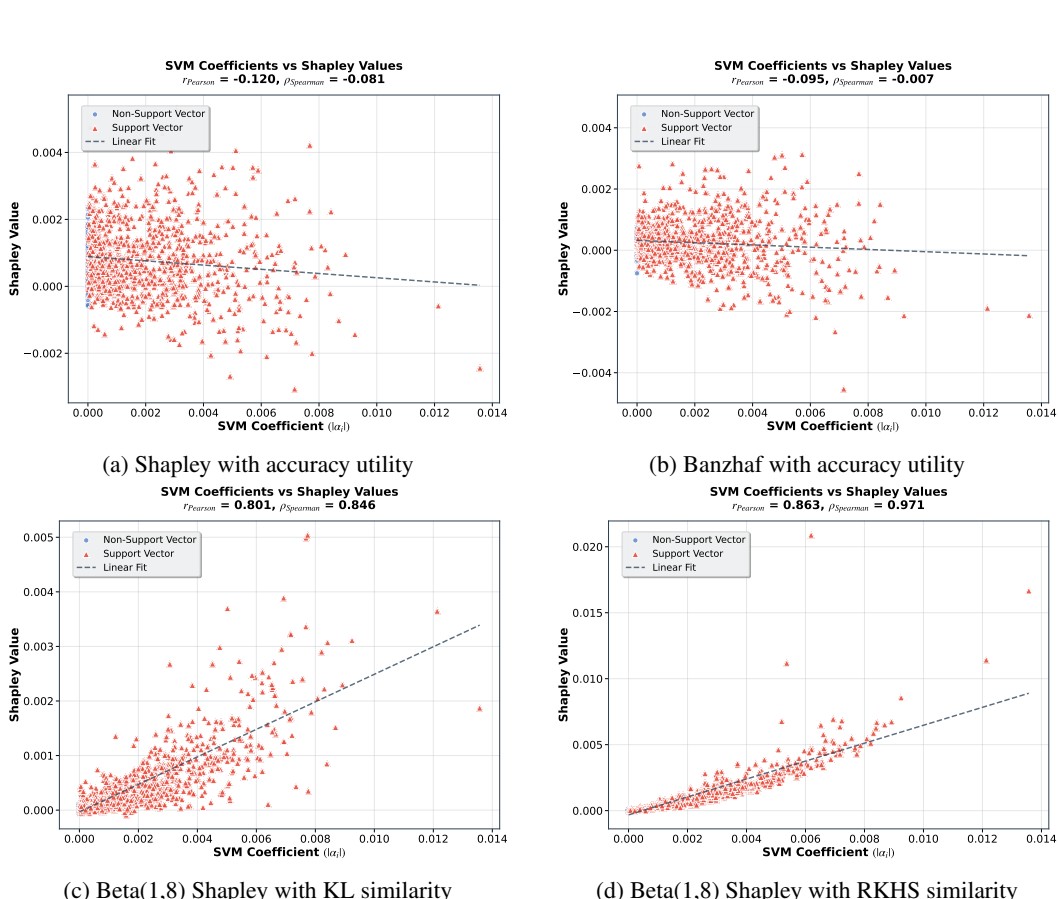

(a) Shapley with accuracy utility

(b) Banzhaf with accuracy utility

(c) Beta(1,8) Shapley with KL similarity

(d) Beta(1,8) Shapley with RKHS similarity

Figure 6: Correlation between $|\alpha_i|$ and data valuation scores on CIFAR-10 (ResNet features). Each point is a training example and color indicates support vector membership. Similarity based utilities show stronger alignment with the decision boundary than accuracy based baselines.

# E  CONCEPTUAL GAP BETWEEN COOPERATIVE GAMES AND DATA VALUATION

**Superadditivity and convex games.**  In a classical cooperative game with player set $N$ the *characteristic function* $v : 2^N \to \mathbb{R}$ is *superadditive* (or *convex*) if

$$v(A) + v(B) \ \leq \ v(A \cup B) \quad \forall A, B \subseteq N, \ A \cap B = \varnothing.$$

Superadditivity guarantees that enlarging a coalition never hurts its overall worth, a property that underpins fairness axioms such as efficiency and the non-negativity of classical Shapley pay-offs.

**Why data valuation breaks superadditivity.**  When $v(\cdot)$ is instantiated as a model-performance metric on a training subset $S$, there is *no* such monotonicity guarantee: a single mislabeled point or a distributional outlier can *reduce* validation accuracy or increase loss, yielding $v(S \cup \{i\}) < v(S)$. Table 1 (Example 2) is a toy demonstration; in practice, flipping 5–10% of labels in CIFAR-10 produces the same effect and drives negative Shapley values.

**Illustrative experiment.**  Figure 7 plots the validation accuracy of a logistic regression model on Wine as we *add* an increasing number of artificially flipped-label samples. The curve clearly dips below the baseline, confirming non-monotone behaviour. Such violations produce *negative* marginal contributions that are legitimate in data valuation but impossible in convex cooperative games.

**Implication for weighting schemes.**  Because non-monotonicity is concentrated in *small* coalitions, our large-subset Beta weighting (Section 4.3) mitigates its destabilising effect, while small-subset-heavy schemes such as Beta Shapley intentionally amplify it for noise-detection tasks.

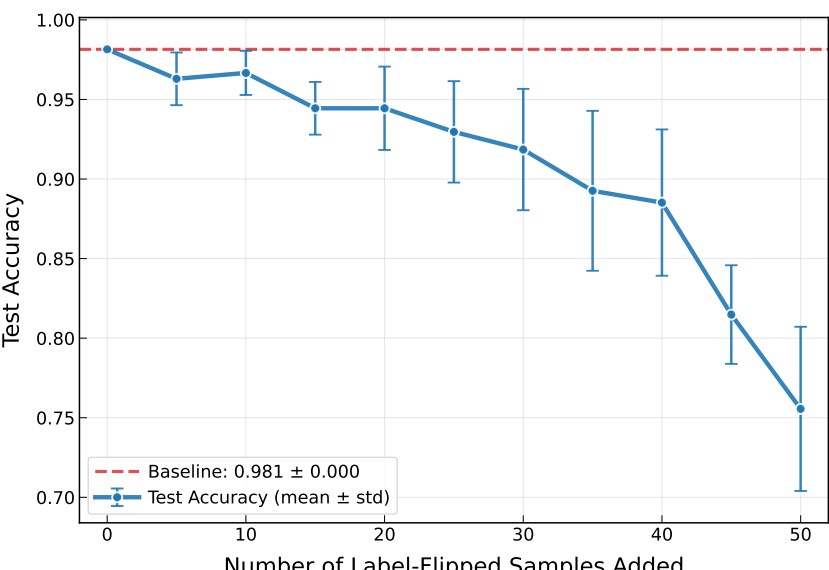

Figure 7: Non-monotonicity of accuracy utility. Test accuracy of a logistic regression model on the Wine dataset degrades as samples with flipped labels are added to the training set

# F  MONTE CARLO ESTIMATION PROCEDURE

---

**Algorithm 1** Monte Carlo Estimation of Beta-weighted Shapley Value

---

**Input:** Dataset $D$, utility function $\mathcal{U}$, Beta parameters $\alpha, \beta$, number of subset sizes $T$, samples per size $M$

**Output:** Estimated Shapley values $\{\mathcal{SV}_i\}_{i=1}^N$

1: Compute discrete weights:

$$\omega_j^{\text{beta}} \leftarrow \frac{\binom{N-1}{j-1} \cdot \text{Beta}(j + \beta - 1, N - j + \alpha)}{\text{Beta}(\alpha, \beta)} \quad \text{for } j = 1, \ldots, N$$

2: **for** each data point $i \in D$ **do**

3:     Initialize $\mathcal{SV}_i \leftarrow 0$

4:     **for** $j = 1$ to $T$ **do**

5:         **for** $m = 1$ to $M$ **do**

6:             Sample subset $S \subset D \setminus \{i\}$ with $|S| = j - 1$

7:             Compute marginal contribution: $\Delta_m \leftarrow \mathcal{U}(S \cup \{i\}) - \mathcal{U}(S)$

8:         **end for**

9:         Compute average: $\overline{\Delta_j(i)} \leftarrow \frac{1}{M} \sum_{m=1}^M \Delta_m$

10:        Update value: $\mathcal{SV}_i \leftarrow \mathcal{SV}_i + \omega_j^{\text{beta}} \cdot \overline{\Delta_j(i)}$

11:     **end for**

12: **end for**

13: **return** $\{\mathcal{SV}_i\}_{i=1}^N$

---

## G    VISUALIZATION OF BETA WEIGHT DISTRIBUTIONS

Figure 8 plots the discrete Beta weights $\omega_j^{\text{beta}}$ for different parameter settings as a function of subset size $j$. The uniform setting Beta(1,1) corresponds to standard Shapley, which treats all subset sizes equally. Beta(8,1) emphasizes small subsets, as used in Beta Shapley (Kwon & Zou, 2022), while Beta(1,8) emphasizes larger subsets, which better align with stable model behavior in our framework.

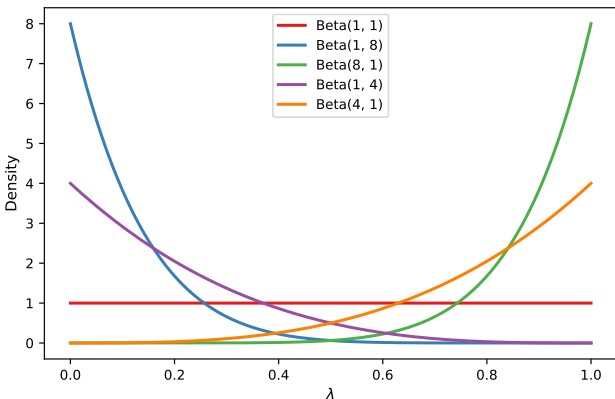

Figure 8: Beta weight distributions over subset size $j$ for different Beta parameters.

# H    GUIDELINES ON CHOOSING BETA WEIGHTS

**Relationship to *Beta Shapley*.**    Both our method and *Beta Shapley* (Kwon & Zou, 2022) belong to the semivalue family that re-weights marginal contributions. The critical difference lies in the *direction* and *purpose* of the weighting: *Beta Shapley* fixes $(\alpha \gg \beta)$ to emphasise *small* coalitions so that individual label-flip errors strongly influence utility—a design that excels at **noise detection**. By contrast, we set $(\alpha \ll \beta)$ to emphasise *large* coalitions where the model behaviour stabilises; combined with similarity-based utility, this suppresses small-subset artefacts and highlights points that consistently shape the global decision boundary (e.g. support vectors).

**Three weighting regimes in practice.**    Empirical evidence across our experiments and prior work suggests a task-dependent choice:

- **Uniform** $(\alpha=\beta=1)$ — maintain all Shapley axioms; suitable when data are balanced and relatively noise-free.
- **Small-subset priority** $(\alpha \gg \beta)$ — useful for *anomaly* or *label-noise* detection where a single sample can flip performance on tiny training sets (the setting studied by *Beta Shapley*).
- **Large-subset priority** $(\alpha \ll \beta)$ — recommended for *structure-preserving* tasks such as support-vector discovery, data pruning, or knowledge distillation, where global decision fidelity is the goal (our setting).

**Selecting $(\alpha, \beta)$ automatically.**    One pragmatic strategy is to treat $(\alpha, \beta)$ as hyper-parameters and choose them via cross-validation on a downstream objective—for example, minimising performance drop in the low-to-high deletion curve (Appendix D.2). In our experiments, $(\alpha, \beta) = (1, 8)$ consistently balanced robustness and discriminative power across datasets.

**Beyond the Beta family.**    Although we adopt the Beta distribution for its flexibility and closed-form weights, other monotone weight families (e.g. truncated power laws) could be used. A systematic exploration of the weight functional space, coupled with variance-minimisation criteria, is left for future work.

## I  COMPUTE ENVIRONMENT AND CODE AVAILABILITY

All experiments were conducted on a dedicated server with the following specifications:

- **CPU Architecture:** x86_64, 2 × AMD EPYC 9754 (128-Core) processors
- **Logical Cores:** 512 (2 sockets × 128 cores × 2 threads per core)
- **Max Clock Speed:** 3.1 GHz (with frequency boost enabled)
- **RAM:** 128 GB
- **Operating System:** Ubuntu 20.04 LTS (64-bit)
- **Virtual Memory Support:** 52-bit physical / 57-bit virtual address space
- **Instruction Set Extensions:** AVX2, AVX-512 (F, BW, VNNI, BF16, etc.), SSE4.2, SHA, AES-NI, and SVM virtualization support

This high-performance configuration enabled large-scale Shapley value estimation with repeated model retraining and subset sampling in a feasible runtime. The implementation to reproduce all experiments in this paper is available at `https://anonymous.4open.science/r/svmsv-3E80/`, and will be released publicly upon acceptance.

