# OpenReview forum: "Rethinking Shapley Value for Data Contribution"
_ICLR.cc/2026/Conference — ICLR 2026 Conference Withdrawn Submission_

### Official Review · Reviewer_9sCH · 2025-10-27

**Soundness:** 3
**Presentation:** 1
**Contribution:** 1
**Rating:** 2
**Confidence:** 4

**Summary:**

An observation is made:  the accuracy combined with the Shapley value  might not well represents how a point from a set drives the model structure. The solution proposed is to change the utility and the semi-value. Compare the model trained on the full dataset against what we  get with a subset of it, then use a semi-value that put more weights on the large coalitions.

**Strengths:**

The paper is timely, and the idea of proposing a specific metrics for the impact on the final model structure is of interest.

**Weaknesses:**

I however regret the lack of analysis as well as too poor arguments.
The presentation is not precise enough with the notion introduced.
Many paragraphs are too vague. Valuations and contributions are claimed to be different, but not defined.
The usage of true and truly reflects this vagueness: "the true X" is not a well formed, unambiguous mathematical statement.
In a well defined context, one should be able to just say "X". This pattern is repeated many times in the paper.
The theoretical properties are mostly direct consequences of the definitions.

**Questions:**

1. Can you provide a definition of contribution?
2. It seems that you suppose that the value is always increasing in a cooperative game, are you? Is it?
3. What is the "true role" of a data point?
4. In example 2 of table 1, C plays the same role as A and B if we exclude the singletons, so can you elaborate on why this is a paradox?
5. Why not using directional derivatives instead of Beta_weighted semi-value, if you are only interested in the final model?
6. Is your redefinition of the KL (with the epsilon) standard?
7. Can you provide more argument on why the final model should be the target? Suppose for instance that half the points have a bad impact on the accuracy, and half the points a good impact, then maybe what we want is to know that the bad points are bad?
Probably what could help me would be a use case for your methodology. What are the downstream tasks you envision for your valuation?

---

> ### Author Response · Authors · 2025-11-19
>
> We thank the reviewer for the insightful question.
>
> Q1: We acknowledge that there is no universally accepted, formal definition of data contribution in the literature. This conceptual ambiguity is, in fact, one of the motivations of our work. To ground the discussion in a verifiable setting, we adopt support vectors in SVMs as a canonical reference for what “structural contribution” means: support vectors are precisely the samples that determine the model’s decision boundary. In other words, we think that any valid contribution measure should ideally recover support vectors in the SVM case. And the traditional method, like the Shapley value, can not do this(fig. 1  and fig. 5)
>
> Q2: No, we do not assume monotonic utility. Machine learning data games are inherently non-monotone, and this is precisely why standard Shapley values can misrepresent the importance of a data point. Our method is designed to handle this non-monotonicity.
>
> Q3: We agree that the notion of a “true role” is not formally defined in the current data valuation literature. This term in our paper was only intended to provide intuition rather than to assert a mathematically unique ground truth.
>
> Since no universal definition exists, we adopt SVMs as a verifiable setting where the structural role of data points is objectively known: the support vectors are exactly those samples that determine the model’s decision boundary. Thus, we use them as a *reference notion of contribution*. Our goal is not to claim an absolute “true role,” but to propose a framework that—under such well-defined conditions—recovers the known influential points. Beyond SVMs, the same idea extends through similarity-based utilities (e.g., KL divergence) to approximate structural influence for general models.
>
> Q4: The paradox in example 2 is that the Shapley value of C is positive. If we adhere to the conclusion of data shapley, we should add C in the adding experiment or retain C in the deletion experiment without hesitation. However, it is obvious that C is harmful for {A,B}.
>
> Q5: The directional derivative characterizes the instantaneous change of a parameter in a certain direction, but the influence of data points is combinatorial: interactions with other points significantly affect the shape of the final solution. semi-values average the marginal values in the union space, systematically integrating interaction terms. Our Beta weighting concentrates quality onto a near-full set, constraining the intuition of "only caring about the final model" while preserving the additive summation of interactions. We will include an experiment(loo vs our method)
>
> Q6: Yes, it is. This follows the standard practice of adding small constants to avoid log⁡0, used in many KL-based objectives such as in variational autoencoders.
>
> Q7:We want to know how much a data point changes the model that is actually deployed. This is why we take the full-data model as the reference and define contribution in terms of its effect on that final model. As for downstream use cases, our primary focus in this submission is data pruning: selecting a small subset of training data that preserves model quality. Empirically, our method achieves this goal more effectively than standard Shapley-based baselines: on several benchmark datasets, removing up to 80% points still maintains high test accuracy.
>
> Prior work such as Data Shapley and Beta Shapley has demonstrated strong performance in tasks like mislabel detection, where the goal is to identify harmful points whose removal improves model accuracy. In these settings, focusing on how “bad” a point is (i.e., its average disruptive effect across many subsets) is entirely appropriate, and these methods excel at capturing such negative influence. Our work, by contrast, targets a complementary problem: identifying *highly beneficial* points that are structurally essential for the final model. In other words, prior methods quantify how bad bad points are, whereas our framework quantifies how good good points are.

---

> > ### Comment · Reviewer_9sCH · 2025-11-26
> >
> > I thank the authors for their answers.
> > However, they did not change my understanding and opinion on the current version of the paper, mostly because the answers are rephrasing of what is already in the paper, I will therefore keep my score.

---

### Official Review · Reviewer_BtCs · 2025-10-29

**Soundness:** 3
**Presentation:** 3
**Contribution:** 3
**Rating:** 6
**Confidence:** 4

**Summary:**

This paper presents a novel framework for measuring data point contribution, developed to address the conflation of data valuation and structural contribution in traditional Shapley value methods. The authors propose a new methodology using a similarity-based utility function and a Beta-weighting scheme, and evaluate its effectiveness on data pruning tasks to identify critical data points.

**Strengths:**

1、The paper's distinction between data valuation and structural contribution addresses a key limitation in traditional Shapley-based methods.
2、The method is grounded in a solid theoretical foundation by combining cooperative game theory with model similarity metrics.
3、The framework's efficacy is demonstrated through well-designed experiments showing superior performance in data pruning tasks.

**Weaknesses:**

1、The validation's exclusive reliance on SVMs creates a risk of circular reasoning, as it benchmarks against support vectors—the very structural elements the method is designed to identify. This narrow empirical scope fails to substantiate the framework's claimed generalizability.
2、The complete omission of a computational cost analysis is a critical oversight for a Shapley-based method. This absence makes it difficult to evaluate the framework's practical scalability and viability for larger-scale problems.

**Questions:**

1、The validation is currently limited to SVMs, which have a clear ground truth in support vectors. Could the authors extend their experiments to include more diverse model architectures, particularly deep neural networks? This would require providing quantitative results and explaining the evaluation protocol used to measure "structural contribution" in the absence of a direct analogue to support vectors.
2、The paper lacks sifficient analysis of computational cost, which is critical for Shapley-based methods. Could the authors provide a complexity analysis or empirical runtime comparisons against baseline methods on identical hardware to assess the framework's practical scalability?
3、The framework is designed to separate "valuation" from "contribution." Could the authors discuss the potential trade-offs of this separation? Specifically, are there scenarios where a data point with high general utility (valuation) might be assigned a low score if it is not structurally critical to the final model, and would this be a desirable outcome?

---

### Official Review · Reviewer_dLM4 · 2025-11-01

**Soundness:** 2
**Presentation:** 3
**Contribution:** 1
**Rating:** 2
**Confidence:** 4

**Summary:**

**Paper**: *Rethinking Shapley Value for Data Contribution* (ICLR under review).
**Core idea**: The paper argues that *data valuation* (average utility across all subsets) is not the same as *data contribution* (a point’s role in shaping the final model). It proposes a semivalue-style framework that (i) replaces accuracy-based utilities with **similarity-based utilities** that align subset models to the full model—specifically, **symmetric KL** over predictive distributions and an RKHS-based functional similarity—and (ii) reweights marginal contributions to **emphasize large coalitions** (e.g., Beta(1,8)) so that scores better reflect stable, full-model behavior. Empirically, the method concentrates value on true support vectors and correlates more strongly with SVM dual coefficients \(|\alpha_i|\), improving SV identification metrics over accuracy-based Shapley/Banzhaf baselines.

**Strengths:**

- **Clear problem framing.** The paper crisply separates *valuation* from *contribution* and explains why small subsets can distort contribution estimates.
- **Well-motivated utilities.** The **symmetric KL** utility captures fine-grained distributional differences between full vs. subset models; the paper gives a local second-order expansion that clarifies its sensitivity. The RKHS option provides a functional alignment view.
- **Principled weighting.** Using **Beta weights** to emphasize large coalitions (e.g., Beta(1,8)) is positioned as complementary to prior “small-subset-focused” semivalues (e.g., Beta Shapley), aligning better with stable full-model behavior.
- **Evidence consistent with goals.** The method concentrates value on *support vectors* and shows stronger correlation with SVM dual coefficients \(|\alpha_i|\), indicating better structural alignment than accuracy-based baselines.

**Weaknesses:**

1) **Critical related work is omitted, overstating novelty.**
The manuscript does not discuss *On the Inflation of KNN Shapley*, a closely related work that **explicitly diagnoses small-subset–induced inflation** and proposes a **subset-size remedy**, which is highly aligned with this paper’s central motivation (mitigating small-subset artifacts). This omission leaves readers without a clear picture of how the present approach differs in assumptions, guarantees, or empirical behavior, and risks **overclaiming novelty**.


2) **Lack of large-scale, deep-learning evaluations.**
Despite solid small/medium-scale evidence, the paper lacks **end-to-end deep-model** or **large-dataset** experiments that stress test scalability and practical value (e.g., modern vision/text benchmarks, pretrained encoders, or full-network training). Without these, it is hard to judge external validity and deployment readiness.

**Questions:**

Same with weakness

---

> ### Author Response · Authors · 2025-11-19
>
> weakness1:
>
> We sincerely thank the reviewer for pointing out this important work. We agree that it is relevant, as both studies identify the instability of uniform Shapley averaging over small subsets. We will add a detailed discussion and citation in Section 2. Conceptually, CKNN-Shapley(On the Inflation of KNN Shapley) proposed that the inflation of KNN-Shapley stems from the unrepresentative nature of the “very small subset”. In contrast, our paper develops a model-agnostic theoretical framework, which uses similarity-based utilities (RKHS or symmetric KL) and Beta-weighted semi-values, to generalize this intuition to arbitrary models with formal properties (Theorems 4.1, 4.3).
>
> weakness2:
>
> We acknowledge this limitation and appreciate the reviewer’s call for broader validation. However, current Shapley-value methods based on subset sampling suffer from exponential computational complexity, making them difficult to apply directly to large-scale models and datasets. Our primary goal in this submission was therefore to establish the conceptual and theoretical foundation of structural contribution and to verify it under controlled, analyzable settings (e.g., SVMs, where the ground-truth structural roles are explicitly known). Because we use beta weighting for subsets of different sizes (with a greater focus on smaller subsets), we will propose a more efficient sampling method in future work.

---

### Official Review · Reviewer_ogdD · 2025-11-01

**Soundness:** 2
**Presentation:** 3
**Contribution:** 2
**Rating:** 2
**Confidence:** 4

**Summary:**

This paper studies the data valuation/contribution problem, where the authors highlight out a conceptual confusion caused by the classical Shapley value concept, that is, the average marginal utility of a data point across all subsets, versus its specific structural contribution in shaping the final model (though this might be a bit of “over-claiming” since recent works on data valuation already accounts for this issue – see detailed comment below).

The authors proposed to replace the accuracy utility function with a similarity-based utility function (RKHS-based function similarity for SVM models, and also model agnostic KL-based similarity for general model probability outputs). In addition, the authors proposed to prioritize larger, more stable subsets’ marginal utilities, which represents a different weighting scheme from standard Data Shaprly and its variants like Beta-Shapley and Banzhaf value. Finally, the authors show experimental results for SVM on different benchmark datasets comparing the proposed method with the classical Shapley value method, and original Beta Shapley value method, and the Banzhaf value method. Experimental results show that the proposed method can effectively assign higher values to support vectors, and keeping valuable points in a removal by ascending values experiment.

**Strengths:**

This paper is generally well written and proposes an interesting research idea, which distinguishes the average marginal utility from the specific structural contribution of a data point. They propose a new variant of Shapley value via similarity-based utility function, together with the Beta-weighting scheme, and show its effectiveness in SVM. The experimental results show that the model can maintain accuracy after removing 80% of the data, which shows some interesting practical implications, such as in the area of data pruning can be explored with the results from this paper.

**Weaknesses:**

In my opinion, there are a few major limitations.

First, the idea of distinguishing data’s value and its “structure contribution” is already there in the literature. This seems to significantly reduce the novelty of this paper’s core motivation “…application has led to a critical, yet often overlooked, conceptual confusion between the value of a data point (its average utility across all subsets) and its specific, structural contribution (its role in shaping the final model)”, and made this statement a little bit over-stated. For example, an example among a few other recent works is “An Instrumental Value for Data Production and its Application to Data Pricing” by [Ai et al., ICML’25], though they call   “structural contribution” the “instrumental value” instead -- however, conceptually these are similar. Moreover, through economic analysis about valuation and contribution, the above ICML  paper arrived at a valuation function that, essentially, put all weight on the largest subset. This paper’s re-weighting approach can be viewed as a “Beta-smoothed” version of that.

Second, the proposed approach is based on some interesting observation, but it lacks solidness and novelty, especially given that the idea of carefully weighting different data subsets with different weights in order to account for some limitations is already extensively explored, and led to a few variants already. It seems to me this is just an additional variant to that literature. In fact, it feels like even the novelty of the re-weighting idea appears somewhat overstated to me. It seems to me, compared to Beta Shapley (Kwon & Zou, 2022),  the only real difference is the parameter choice (beta > alpha instead of alpha > beta), which flips the bias from small to large subsets (though I know this paper also proposed a different valuation function).

Third, the experiments are limited to support the claim that the method is model-agnostic. Specifically, results are only shown for SVMs and one logistic regression test on the Wine dataset. Besides the neural network models omitted by authors due to “prohibitive retraining (Line 466)”, there are no results on other common simple models like random forests or even decision trees, which are easy to retrain and help demonstrate contribution in more diverse settings. Even with logistic regression, experimental results on more than just the Wine dataset would make the model-agnostic claim stronger. This leads to significant concern about the broad applicability of the proposed approach, especially that I do not see any intuition that it would work for other ML models, particularly today’s large auto-regressive model or diffusion models, where it seems every piece of data matters.

Forth, some of the theoretical results are relatively straightforward. For instance, Theorem 4.3 seems straightforward – if we have larger value for all subsets with size at most m, then putting “sufficiently large” weights on these size surely will make the averaged value larger.  It seems to me a more interesting statement might be, for what parameter sets of your chosen Beta distribution, it will lead to larger value for i.

Fifth, another weakness is, in the experiments, one major missing data valuation method is the leave-one-out (LOO) baseline, which directly estimates a data point’s marginal contribution when removed from the full training set. This is an extreme case of large subset emphasis that aligns with the authors’ goals. Also this baseline was explicitly included in the original Beta Shapley work (Kwon & Zou, 2022) as a comparison, but it is entirely absent here, with no discussion or justification for its exclusion.

**Questions:**

How does varying these two parameters (i.e., other values other than Beta(1, 8) but keeping beta > alpha) affect the algorithm’s performance? It would be helpful to provide some results when varying these two parameters.

How do all other algorithms perform on the sorted Shapley value experiment (i.e., Figure 1) of each data point? It seems only Standard Shapley and Beta(1,8)-kl are presented. Also, is there an explanation for the tail elements always being support vectors?

---

### Note · Authors · 2025-11-27

I have read and agree with the venue's withdrawal policy on behalf of myself and my co-authors.